

# Identifying model violations under the multispecies coalescent model using P2C2M.SNAPP

Drew J. Duckett[1], Tara A. Pelletier[2] and Bryan C. Carstens[1]

[1] Department of Evolution, Ecology, and Organismal Biology, The Ohio State University, Columbus, OH, USA
[2] Biology Department, Radford University, Radford, VA, USA

## ABSTRACT

Phylogenetic estimation under the multispecies coalescent model (MSCM) assumes all incongruence among loci is caused by incomplete lineage sorting. Therefore, applying the MSCM to datasets that contain incongruence that is caused by other processes, such as gene flow, can lead to biased phylogeny estimates. To identify possible bias when using the MSCM, we present P2C2M.SNAPP. P2C2M.SNAPP is an R package that identifies model violations using posterior predictive simulation. P2C2M.SNAPP uses the posterior distribution of species trees output by the software package SNAPP to simulate posterior predictive datasets under the MSCM, and then uses summary statistics to compare either the empirical data or the posterior distribution to the posterior predictive distribution to identify model violations. In simulation testing, P2C2M.SNAPP correctly classified up to 83% of datasets (depending on the summary statistic used) as to whether or not they violated the MSCM model. P2C2M.SNAPP represents a user-friendly way for researchers to perform posterior predictive model checks when using the popular SNAPP phylogenetic estimation program. It is freely available as an R package, along with additional program details and tutorials.

# INTRODUCTION

Alleles that are shared across taxa present a formidable challenge to phylogenetic inference. Species tree inference methods were introduced in an attempt to infer phylogeny without the potentially confounding effects caused by ancestral alleles that were shared across operational taxonomic units (OTUs) (*Maddison, 1997*; *Carstens & Knowles, 2007*). Since the biological mechanisms that lead to this process (i.e., incomplete lineage sorting) commonly occur at shallow levels of phylogenetic divergence, species trees have largely (but not exclusively; *Prum et al., 2015*) been applied near the species boundary, and often in clades where species limits are not entirely clear (*Satler, Carstens & Hedin, 2013*). Such applications of the species tree model make the implicit assumption that alleles shared across lineages result from incompletely sorted ancestral polymorphism, even though gene flow is possible in closely related taxa. While gene flow was once considered rare above the species level (at least in animals), recent investigations have suggested that it is more

Corresponding author
Bryan C. Carstens,
carstens.12@osu.edu

common than previously recognized (e.g., snowshoe hares: *Melo-Ferreira et al. (2014)*, chipmunks: *Sullivan et al. (2014)*, bears: *Kumar et al. (2017)*, and *Myotis* bats: *Morales et al. (2017)*).

Given that gene flow has been shown to bias estimates of both topology and branch lengths when it is not accounted for in a phylogenetic analysis (*Eckert & Carstens, 2008*; *Leaché et al., 2013*), evolutionary biologists should (at the least) consider the possibility that gene flow has interfered with phylogeny estimation, particularly when inferring phylogenies from closely related species where reproductive isolation may not be complete. One approach is to look for evidence of gene flow in the data, for example, by searching for alleles that are shared across non-sister taxa because such alleles are more likely to result from gene flow than coalescent processes. However, this is likely to be a laborious process, particularly in genomic datasets, and gene flow can be easily missed in studies that do not analyze data from all possible hybridization/introgression events. It is considerably more efficient to utilize statistical methods, such as posterior predictive simulation, that seek to determine whether a given dataset violates the model assumptions of the phylogenetic analysis (*Goldman, 1993*; *Reid et al., 2014*).

Posterior predictive approaches have been developed for several types of phylogenetic models, including models of sequence evolution (*Huelsenbeck et al., 2001*; *Brown, 2014b*), species delimitation (*Barley & Thomson, 2016*; *Barley, Brown & Thomson, 2018*), and species tree estimation (*Reid et al., 2014*). The basic approach is to (i) draw parameter values from the posterior distribution, (ii) simulate new datasets using these parameter values under the model assumed by the analysis, (iii) analyze the simulated data to generate posterior predictive distributions, and (iv) calculate and compare summary statistics from either the empirical data or the posterior distribution to the posterior predictive distribution. Analytical models that represent a good fit for the empirical data should produce summary statistics values that fall within the distribution of values estimated under the correct model with posterior predictive datasets (*Brown, 2014b*). Recently, posterior predictive checks have been incorporated into an R package (Posterior Predictive Checks of Coalescent Models (P2C2M): *Gruenstaeudl et al., 2016*) for the multispecies coalescent model (MSCM) framework. P2C2M was designed to easily allow users to perform posterior predictive analyses, but the program uses the species tree inference package *BEAST which is intended for smaller, sub-genomic data sets (*Heled & Drummond, 2010*). Here, we expand P2C2M to the genomic era so that it can be used to conduct posterior predictive checks using single nucleotide polymorphisms (SNPs) in the SNAPP implementation of the MSCM (*Bryant et al., 2012*).

## MATERIALS AND METHODS

### Pipeline

The posterior predictive simulation framework for SNAPP (P2C2M.SNAPP) has been implemented as an R package (*R Core Team, 2018*), with detailed program settings described in the package documentation and tutorial. P2C2M.SNAPP differs from the original P2C2M in the input datatype (sequence data in the original versus SNP data in the SNAPP version) and consequently the summary statistics used to compare empirical and
posterior predictive datasets. User input to P2C2M.SNAPP includes the SNAPP .xml formatted input file, the posterior distribution of species trees and log file from a SNAPP analysis, and a metadata text file containing the number of SNPs used, an estimated mutation rate, and the number of samples per group. Importantly, P2C2M.SNAPP assumes users have properly performed SNAPP species tree estimation analysis, including selecting the proper priors for their data and study system and checking for Markov chain convergence. Because P2C2M.SNAPP relies on the posterior distribution of species trees, users should retain at least 100 trees in the posterior distribution to sample from. P2C2M.SNAPP proceeds as follows: (i) it samples, either uniformly or at random, a user-specified number of species trees from the posterior distribution, (ii) extracts taxonomic relationships and branch lengths from each tree, and (iii) for each tree sampled from the posterior, it simulates a posterior predictive dataset under the MSCM using fastsimcoal2, a user-specified number of simulations (*Excoffier et al., 2013*) and the parameters extracted from the metadata text file (Fig. 1). Posterior predictive datasets are converted to SNAPP .xml files, and users conduct SNAPP analyses on each posterior predictive dataset using the .xml file output by P2C2M.SNAPP. Prior distributions and Markov chain parameters for the posterior predictive SNAPP analyses are recycled from those used in the original SNAPP analysis in order to maintain consistency. Given the intense computational requirements of SNAPP, generation of the posterior predictive species tree distributions is best conducted using parallel computation. Example scripts for automating SNAPP analyses are included with the tutorial (http://www.github.com/P2C2M/P2C2M_SNAPP). The results of SNAPP analyses on the posterior predictive datasets (i.e., SNAPP .xml files, posterior species tree distributions, and log files) are subsequently used as input for the second stage of the P2C2M.SNAPP analysis, where summary statistics from the posterior and posterior predictive datasets are calculated and compared to identify model violations.

## Summary statistics

Generally, summary statistics used in posterior predictive checks fall into two categories: data-based, which compare the empirical and posterior predictive datasets themselves, and inference-based, which compare the inferences produced by analyzing the empirical and posterior predictive datasets (*Brown, 2014a*; *Barley & Thomson, 2016*). Inference-based statistics can provide more insight as to whether a model violation affects the end result (e.g., the estimated species tree), but can also be more computationally difficult because posterior predictive datasets need to be analyzed with the same methods as the posterior (i.e., species trees need to be estimated with SNAPP). In contrast, data-based statistics do not determine the effect a model violation has on the inference, but are usually computationally efficient. Both data-based and inference-based summary statistics were evaluated to determine which statistic identified model violations to the MSCM with the highest accuracy. Data-based statistics included several based on a fixation index ($F_{ST}$), and inference-based statistics included tree metrics based on Robinson–Foulds or Kuhner–Felsenstein tree distances, and the mean and standard deviation of tree likelihoods. $F_{ST}$ is a commonly used metric for measuring the amount of population

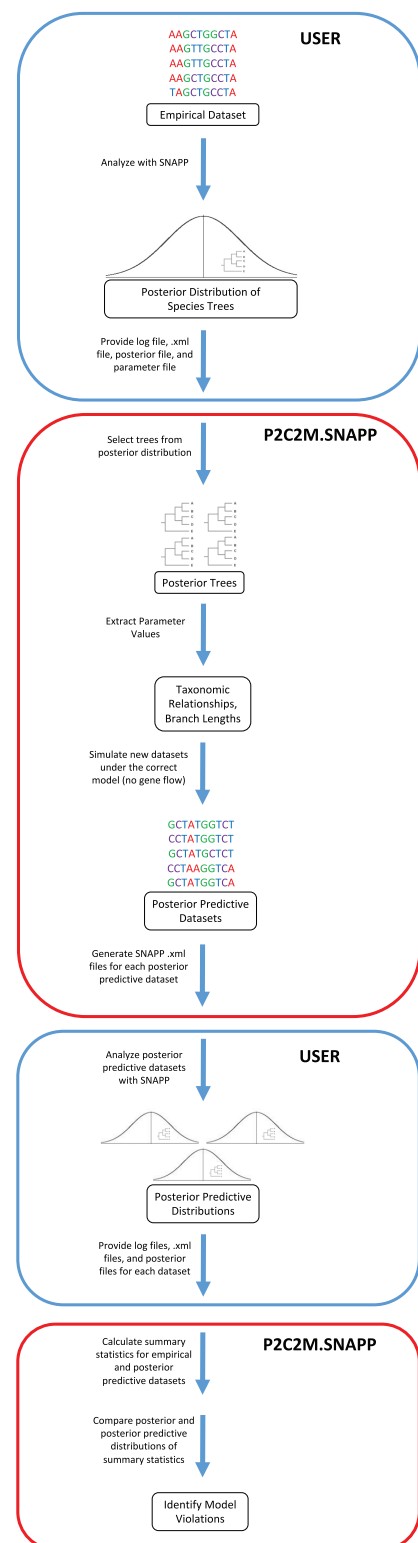

**Figure 1 Workflow of the P2C2M.SNAPP pipeline.** Blue arrows represent the path of the data. Steps outlined in blue are those performed by the user and steps outlined in red are performed by P2C2M. SNAPP. The workflow proceeds from the top of the figure.

structure, and the value ranges from 0 to 1, with populations becoming more structured as $F_{ST}$ approaches 1 (*Wright, 1949*). Therefore, lineages exchanging genes should exhibit lower $F_{ST}$ values because they will share alleles. Pairwise $F_{ST}$ was calculated across all loci in the KRIS package (*Chaichoompu et al., 2018*). $F_{ST}$ summary statistics included mean $F_{ST}$, range of $F_{ST}$ and an $F_{ST}$ outlier test. For the mean and range $F_{ST}$ statistics, the summaries are calculated for each posterior predictive dataset and the empirical dataset. Similar to a two-tailed posterior predictive *p*-value (*Brown, 2014a*; *Barley, Brown & Thomson, 2018*), a *p*-value is calculated by counting the number of posterior predictive datasets with summary statistic values falling above and below the empirical value, multiplying the lesser of these values by two (to emulate a two-tailed test), and then dividing by the total number of posterior predictive datasets. We consider *p*-values less than $\alpha = 0.05$ to indicate a model violation. The $F_{ST}$ outlier test was conducted by calculating the average difference between empirical and simulated values for each pairwise comparison, and then conducting an outlier test using the function boxplot.stats in the grDevices package (*R Core Team, 2018*). Since we consider any detected outlier to indicate a model violation, the pairwise outliers identified by this approach can be used to identify lineages exchanging genes.

Two tree distance metrics were also examined, one that considers topology only and one that considers topology and branch lengths. The Robinson–Foulds distance compares the topology between two phylogenetic trees, with values ranging from 0 (no topology difference) to 1 (completely different topologies) (*Robinson & Foulds, 1981*). High rates of gene flow can influence topology estimation and result in an errant clade consisting of two lineages that are not closely related but that share alleles due to gene flow. However, it may be more likely that gene flow may mislead the estimation of branch lengths even if the underlying topology is correct. Therefore, a tree distance metric incorporating branch length differences as well as topology may prove to be a useful summary statistic for comparing empirical and posterior predictive datasets. One such metric is the Kuhner–Felsenstein distance, which also calculates values between 0 (no difference between trees) and 1 (high difference between trees) (*Kuhner & Felsenstein, 1994*). Both tree distance metrics were calculated using the ape package (*Paradis, Claude & Strimmer, 2004*). If posterior trees were estimated from a dataset that violates the MSCM model, we expect that these trees will have large tree distances when compared to posterior predictive trees simulated under the correct model (MSCM). Additionally, as all posterior trees reflect similar processes in the empirical dataset, we expect that tree distances among trees in the posterior under a model violation will be less than distances between the posterior and posterior predictive trees. Therefore, for the tree distance metrics, 1,000 comparisons were performed between random trees sampled from the original SNAPP posterior distribution of species trees to create a null distribution. Then 100 random trees from the posterior predictive distribution were compared to the posterior tree they were simulated from, and this was repeated for each posterior predictive dataset. A *p*-value was calculated by counting the number of posterior predictive to posterior tree comparisons falling above the 95% null distribution (values below the 95% null distribution represent high similarity between posterior and posterior predictive datasets,

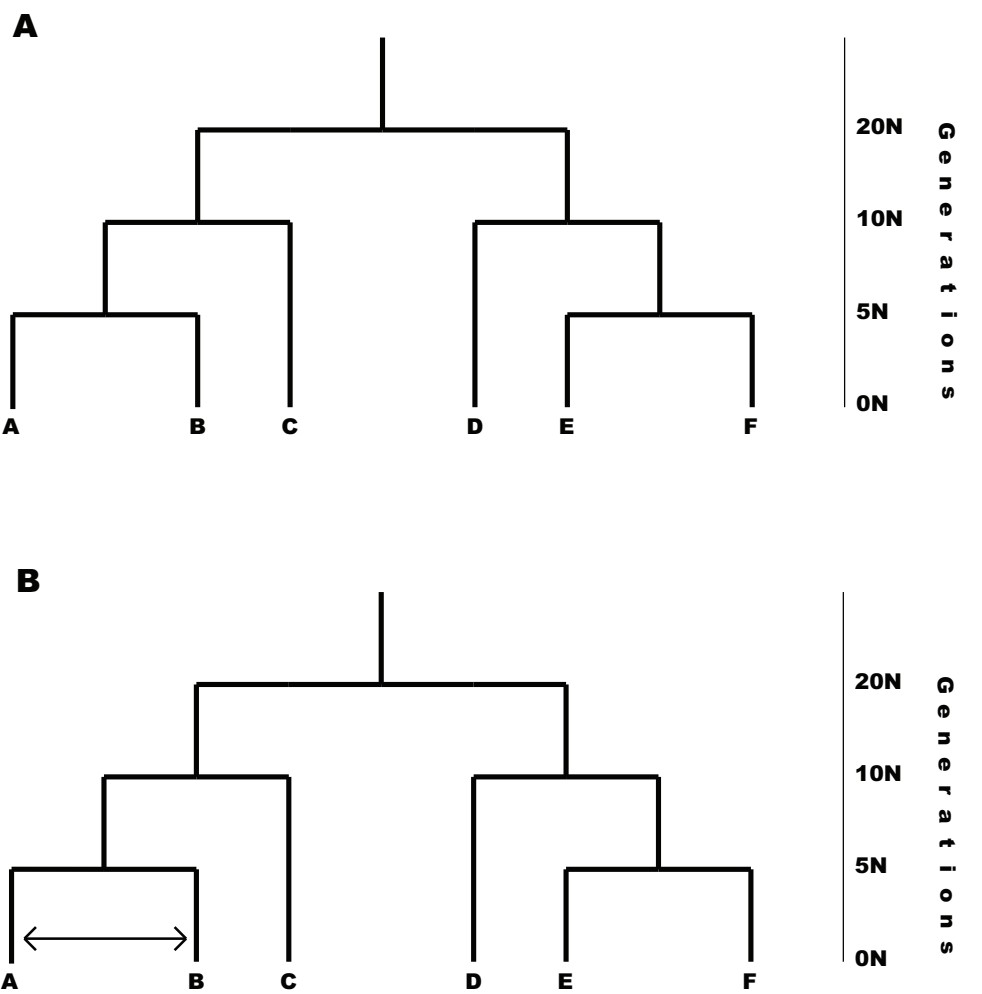

**Figure 2 Models used in simulation testing.** (A) MSCM model used for simulation testing. (B) Example of the MSCM+*m* model that includes gene flow violating the MSCM model implemented in SNAPP. The amount of gene flow and taxa exchanging genes were randomly selected for each simulation replicate.

and thus are not useful for detecting violations), and then dividing by the total number of comparisons. We consider *p*-values greater than α = 0.05 to indicate model violations. Finally, because it is likely more difficult to estimate trees with high probability under an incorrect model, we examined the mean and standard deviation of tree likelihoods as calculated from SNAPP output. The evaluation of the likelihood statistics follows that of the mean and range $F_{ST}$ statistics, described above.

## Testing

P2C2M.SNAPP was tested by simulating data under the MSCM and via a second simulation under the MSCM with gene flow (i.e., MSCM+*m*; Fig. 2). One hundred replicates were performed under each model. Note that the MSCM+*m* model is a clear violation of the underlying coalescent model that is incorporated into SNAPP because an appreciable portion of the shared polymorphism results from gene flow. All simulations

were based on 2,000 SNPs, six species with two individuals sampled per species, an effective population size ($N_e$) of 100 K individuals, and a symmetric topology with speciation event times of 5 N, 10 N and 20 N generations. The number of SNPs simulated is lower than many empirical data sets, but it allows SNAPP analyses to proceed in less time and should represent a conservative test of the ability of P2C2M.SNAPP to detect model violations because the performance of SNAPP generally improves with additional data (*Bryant et al., 2012*). The MSCM+*m* model was designed as a secondary contact scenario, with gene flow between two lineages starting at 2.5 N generations in the past and continuing until the present. Both the species experiencing gene flow and the rate of gene flow were selected at random, with the rate of gene flow having a uniform prior distribution between 0.5 and 5 migrants per generation. Simulations were performed using fastsimcoal2 (*Excoffier et al., 2013*) and simulated datasets were converted to SNAPP .xml files using custom Python scripts (http://www.github.com/P2C2M/P2C2M_SNAPP). SNAPP analyses were conducted using the following parameters: a gamma prior on the rate of species divergence (lambda) under the Yule speciation prior with α = 2 and β = 200, a gamma rate prior on ancestral effective population sizes (theta), mutation rate of $\mu = \nu = 1.0$, and a Markov chain of 1 M steps with 100 K burn-in steps and sampling every 1 K steps. In order to evaluate the summary statistics, the number of correct inferences, false positives, and false negatives were calculated for each model (200 total) using the posterior and posterior predictive distributions from the SNAPP analyses. False positives are defined as datasets simulated under the MSCM that were indicated as model violations by P2C2M.SNAPP. Conversely, false negatives are defined as datasets simulated under the MSCM+*m* model that were not detected as model violations by P2C2M.SNAPP. Mathews Correlation Coefficient (MCC; *Matthews, 1975*) was also calculated for each summary statistic with the R package mltools (*Gorman, 2018*). The MCC takes into account false negatives and positives while measuring how well a binary classifier performs, in this case whether a summary statistic correctly classifies a dataset. The coefficient ranges from −1 to 1 with −1 indicating the classifier is completely wrong and 1 indicating it is completely correct. Additionally, pairwise $F_{ST}$ outliers were compared to the MSCM+*m* simulation parameters to assess if the statistic could identify the species exchanging genes to cause model violations. Finally, *p*-values for each simulation were plotted against gene flow to identify any trends between the level of gene flow and summary statistic performance.

## RESULTS

P2C2M.SNAPP requires about 5 min on an average laptop (2.6 GHz Intel Core i5, 8 GB RAM) to generate posterior predictive datasets at the beginning of the pipeline and to evaluate summary statistics in order to identify model violations at the end of the pipeline. However, the entire pipeline requires a considerable amount of time due to the demands of the SNAPP program itself. For example, each replicate of our simulation testing required 300–450 CPU hours on the Pitzer cluster (28 cores and 112 GB RAM) at the *Ohio Supercomputer Center (2018)*. While this is clearly not an analysis that users would likely conduct on a laptop computer, the time required for users to analyze their data using

**Table 1 Results of simulation testing.** Results include all simulations with both the MSCM and MSCM+*m* models. False positives are datasets simulated under the MSCM model which P2C2M.SNAPP classified as a model violation. False negatives are datatsets simulated under the MSCM+*m* model that P2C2M.SNAPP classified as not violating the model implemented in SNAPP.

| Statistic | True positives | True negatives | False positives | False negatives | Matthews correlation coefficient (MCC) |
|---|---|---|---|---|---|
| Average pairwise $F_{ST}$ (FSTA) | 66 | 0 | 100 | 34 | −0.45 |
| Range of pairwise $F_{ST}$ (FSTR) | 81 | 0 | 100 | 19 | −0.32 |
| $F_{ST}$ outlier test (PFST) | 3 | 88 | 12 | 97 | −0.17 |
| Kuhner–Felsenstein distance (KF) | 100 | 0 | 100 | 0 | 0.00 |
| Robinson–Foulds distance (RF) | 0 | 100 | 0 | 100 | 0.00 |
| Mean of maximum likelihood (MLM) | 84 | 0 | 100 | 16 | −0.29 |
| Standard deviation of maximum likelihood (MLSD) | 71 | 95 | 5 | 29 | 0.68 |

P2C2M.SNAPP is still likely to be less than the time required to collect the samples, generate the sequencing libraries, and conduct the bioinformatics.

There was a dramatic difference across summary statistics in the ability of P2C2M. SNAPP to identify model violations due to gene flow (Table 1). The mean and range of pairwise $F_{ST}$ values correctly classified datasets in only 33% and 41% of simulations respectively (MCC equals −0.45 and −0.32 respectively; Fig. 2). Each of these statistics exhibited a large number of false positives in which a model violation was detected in a dataset that was simulated under the assumptions of the MSCM. While the pairwise $F_{ST}$ outlier test classified 46% of datasets correctly, the majority of misclassifications were false negatives (MCC equals −0.17). Additionally, we examined the ability of the pairwise $F_{ST}$ outlier test to identify the OTUs exchanging genes in the MSCM+*m* datasets. As the statistic only identified 3% of true model violations, there were very few datasets to test. The pairwise $F_{ST}$ outlier test did not correctly identify the OTUs exchanging genes in any of the datasets. Each tree statistic correctly classified only around half of the datasets, with a high number of false negatives when using the Robinson–Foulds distance and a similar number of false positives when using the Kuhner–Felsenstein distance (MCC equals 0 for each). Similarly, evaluations by the mean tree likelihood statistic were split evenly between correct inferences and false positives (MCC equals −0.29). Our results identified one statistic that performed well. The standard deviation of tree likelihoods correctly classified 83% of simulated datasets, with 14% false negatives and 3% false positives (MCC equals 0.68). Only two summary statistics showed a trend between the rate of gene flow and the *p*-value of posterior predictive checks (Fig. 3). For the range of pairwise $F_{ST}$ and mean of tree likelihoods, *p*-values decreased as the rate of gene flow increased.

## DISCUSSION

While it has been known for some time that model violations can degrade the accuracy of phylogenetic estimation (*Huelsenbeck et al., 2001*; *Eckert & Carstens, 2008*, *Leaché et al., 2013*; *Brown, 2014a*; *Reid et al., 2014*; *Barley & Thomson, 2016*;

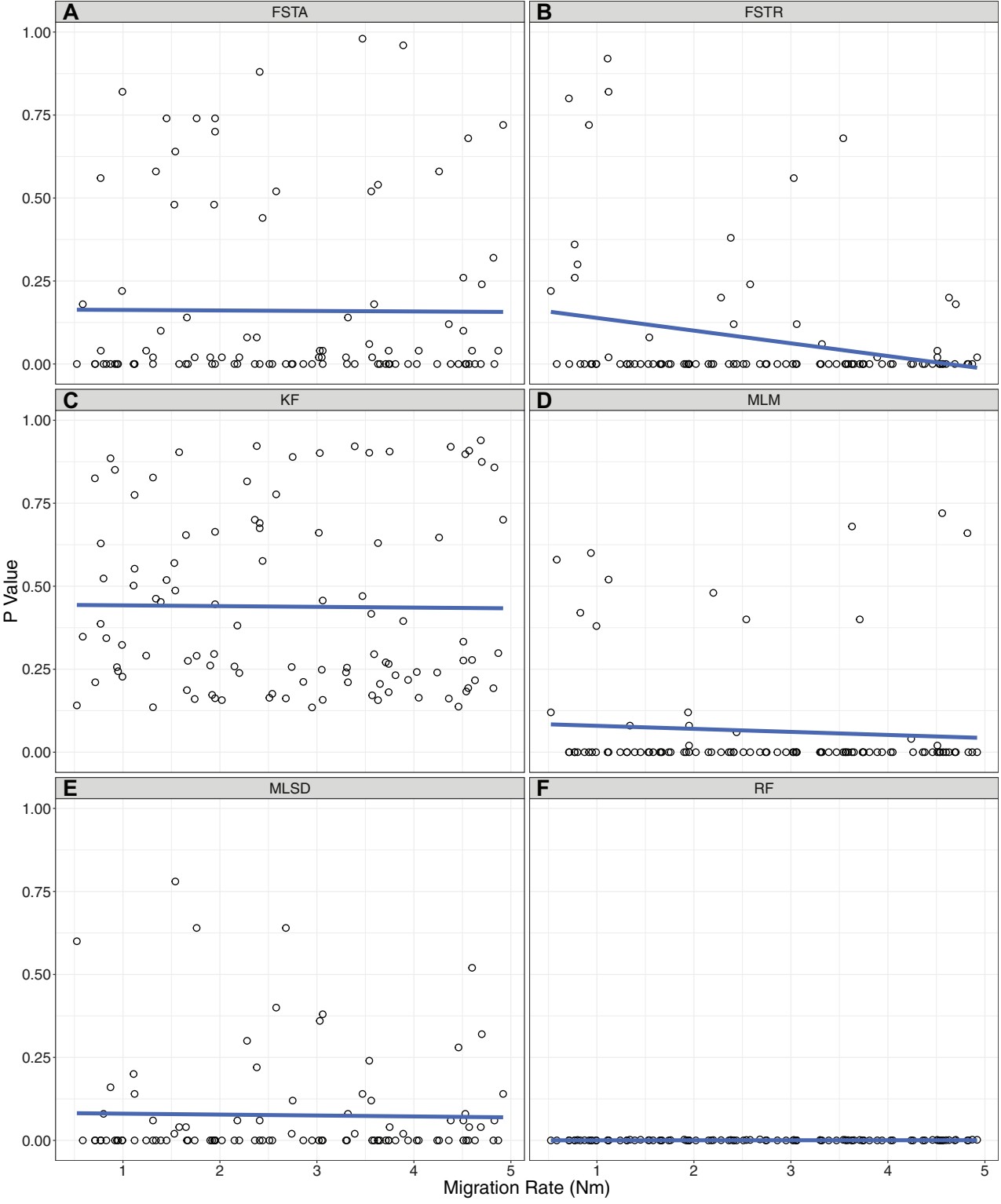

**Figure 3 Correlations between the level of gene flow and the ability of each summary statistic to identify model violations.** The *p*-value for each MSCM+*m* simulation is plotted against the amount of gene flow simulated with that dataset. (A) FSTA: average pairwise $F_{ST}$. (B) FSTR: range of pairwise $F_{ST}$. (C) KF: Kuhner–Felsenstein distance. (D) MLM: Mean of the maximum likelihood of posterior trees. (E) MLSD: standard deviation of the maximum likelihood of posterior trees. (F) RF: Robinson–Foulds distance.

*Barley, Brown & Thomson, 2018*), few studies explore possible violations inherent in their datasets to the phylogenetic model used in the analysis (*Morales et al., 2017*; *Diaz et al., 2018*; *Richards et al., 2018*). Apart from the computational demands of the SNAPP analyses, P2C2M.SNAPP represents a user-friendly and reasonably accurate method for identifying violations of the MSCM. The package and tutorial, including examples for running analyses, are available on the P2C2M GitHub page (https://github.com/P2C2M/P2C2M_SNAPP).

Our simulation testing indicates that the standard deviation of tree likelihoods is useful in identifying datasets that contain SNP patterns resulting from gene flow between lineages, a clear violation of SNAPP's analytical model. This statistic is likely useful because datasets that violate the MSCM model will be more difficult to estimate and may exhibit posterior distributions with poor convergence. Methods examining the variance within and between posterior and posterior predictive datasets have previously proven useful for posterior predictive checks of Bayesian phylogenetic models (*Gelfand & Ghosh, 1998*; *Lewis et al., 2014*). Users of P2C2M.SNAPP should focus on the standard deviation of tree likelihoods when assessing their datasets. Although higher rates of gene flow should result in a more egregious model violation, it does not appear to be the case that model violations are easier to detect under scenarios with high rates of gene flow. Two summary statistics (range of pairwise $F_{ST}$, mean of tree likelihoods) exhibit an inverse correlation between gene flow and the resulting *p*-value, but both exhibited a high rate of false positives which makes them a poor choice for use in posterior predictive checks. While this relationship does not hold for the standard deviation of tree likelihoods, the statistic is able to detect model violations equally well across a range of migration rates.

Several statistics were much less useful than we expected them to be. Although the tree distance metrics are conceptually simple, their poor performance may be explained by the reliance of posterior predictive simulation on the empirical phylogeny. Because the posterior predictive data sets are simulated from the empirical phylogeny estimates, inaccuracies in topology and branch lengths of the empirical phylogeny due to gene flow are translated into inaccurate topology and divergence times in the posterior predictive simulations. The result is similar, but inaccurate phylogeny estimates for each data type. $F_{ST}$ is a popular metric in population genetics for examining population structure and gene flow, but may not be applicable to phylogenetic analyses due to fixed differences among lineages. It is possible that including more samples per lineage may increase the usefulness of $F_{ST}$ because more shared polymorphism may be evident, but this may be unfeasible due to the computational requirements of SNAPP. Summary statistics such as $F_{ST}$ are appealing because they can be computed from the posterior predictive datasets without additional SNAPP runs, but many existing statistics were developed for population genetic applications. Summary statistics such as the number of shared or private alleles may be useful. Additionally, the calculation of effect sizes could be beneficial to users because it provides information regarding the degree to which model violation has influenced their results (*Brown, 2014a*). Our simulation design investigated a relatively recent diversification scenario because the presence of gene flow is likely to occur when lineages have not become completely reproductively isolated. However, if gene flow occurs in older systems, it should

presumably be easier to differentiate from incompletely sorted ancestral polymorphism and thus more easily recognized. Finally, other processes, such as natural selection, also violate the MSCM model, and these additional model violations may potentially be detectable using the posterior predictive framework implemented in P2C2M.SNAPP, but further research is necessary to identify summary statistics that can detect these violations.

While the detection of a model violation can have implications for the interpretation of a phylogeny estimate, a model violation does not render the data useless. Minimally, researchers should acknowledge the model violation and temper their interpretation of the patterns evident in the phylogeny. Specifically, the possibility that a model violation may have confounded topology estimates or, more likely, biased branch length/divergence time estimates should be addressed. More preferably, researchers should conduct additional analyses to examine the cause of the model violation, as such violations indicate interesting evolutionary processes not accounted for by the MSCM model. In the case of gene flow, model violations can indicate unknown hybridization among OTUs, and lead to the collection of population-level data that can be analyzed using methods such as Migrate-$n$ (*Beerli & Felsenstein, 2001*) or Bayesass (*Wilson & Rannala, 2003*). Finally, many recently developed models attempt to infer gene flow and phylogeny under the MSCM for small numbers of lineages (e.g., IMa3: *Hey et al., 2018*; PhyloNet: *Wen et al., 2018*; SpeciesNetwork: *Zhang et al., 2018*). Model violations identified by P2C2M.SNAPP are likely to point researchers to additional analyses that will enable them to understand the history of their focal system.

## CONCLUSIONS

Here we present a new R package for assessing model violations in the species tree estimation program SNAPP. The package uses posterior predictive simulations to identify model violations, and is successful in testing with simulated datasets. P2C2M.SNAPP is the newest addition to a small suite of user-friendly programs for conducting posterior predictive checks (*Gruenstaeudl et al., 2016*). Due to the proven benefit of model checking for phylogenetic analyses, we recommend researchers make posterior predictive checks a routine step in estimating phylogenies.

## ACKNOWLEDGEMENTS

We would like to thank the Carstens laboratory for constructive conversations about this project and providing feedback on the manuscript before submission. We also thank Remco Bouckaert for answering questions about the SNAPP program and the Ohio Supercomputer Center for early access to the Pitzer computing cluster.

### Funding

This work was funded by a grant from the National Science Foundation (DBI-60057056) to Bryan C. Carstens. The funders had no role in study design, data collection and analysis, decision to publish, or preparation of the manuscript.

## Grant Disclosures
The following grant information was disclosed by the authors:
National Science Foundation: DBI-60057056.

## Competing Interests
The authors declare that they have no competing interests.

## Author Contributions

- Drew J. Duckett conceived and designed the experiments, performed the experiments, analyzed the data, prepared figures and/or tables, authored or reviewed drafts of the paper, and approved the final draft.
- Tara A. Pelletier conceived and designed the experiments, analyzed the data, prepared figures and/or tables, authored or reviewed drafts of the paper, and approved the final draft.
- Bryan C. Carstens conceived and designed the experiments, prepared figures and/or tables, authored or reviewed drafts of the paper, funded work, and approved the final draft.

## Data Availability
The raw data and the code are available as a Supplemental File.

P2C2M_SNAPP is available at GitHub: https://github.com/P2C2M/P2C2M_SNAPP and the Supplemental Files are available at https://carstenslab.osu.edu/software.html.

## Supplemental Information
Supplemental information for this article can be found online at http://dx.doi.org/10.7717/peerj.8271#supplemental-information.

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
