# Peer review of "Identifying model violations under the multispecies coalescent model using P2C2M.SNAPP"

_PeerJ, doi:10.7717/peerj.8271_

## Round 0.1 · original submission · Minor Revisions

All reviewers commented favourably regarding the overall utility of the R package described in your submission, and mostly focused on identifying improvements to the manuscript, and opportunities for future work. Below I group together specific revision comments and key points that should be addressed in your revised submission.

Specific points
1. General background/Introduction.
line 50. "While gene flow was once considered rare above the species level (at least in animals), recent investigations have suggested that it is more common than previously recognized". As a layreader, it would be helpful if you mentioned the taxa that have been recently identified to be subject to this rather than simply citing the studies.

i. line 57. "consider the possibility that gene flow has interfered with phylogeny estimation, particularly in investigations near the species level.". Perhaps again due to my unfamiliarity with the specific field, I am unsure what you actually mean by the phrase 'near the species level' - an explicit description would be less ambiguous here.

ii. line 212-213. You provide citations stating that model violation is a problem. You state, however "few studies explore possible violations.." - but provide no citations for these 'few studies' - if there are in fact no studies to your knowledge then it is better to say that - otherwise please include relevant citations here - or even better, mention them in the introduction.


2. You state that this work adds to a 'small group of implementations' of the P2C2M methodology. As recommended by R3 it would be useful to more clearly describe the origins of the methodology, and also explain any significant differences between the P2C2M protocol as implemented in P2C2M.SNAPP vs the canonical R implementation: P2C2M.


3. Reviewer 3 notes that readers unfamiliar with summary statistics computed by the package need more background to interpret these data. They also note that later on in the discussion little explanation is provided as to why the majority of statistics perform so poorly. Reviewer 1 echoes this, and asks if other summary statistics could be computed - particularly those as easy to compute as Fst. R3 in particular thought that a clearer discussion as to why Fst was chosen amongst the various metrics available would be helpful.

i. line 110. F_st should be cited when first mentioned. As R3 also notes, a brief description could usefully be included here for lay readers.

4. Figure 1 should be revised to emphasise the steps performed by the package and what information must be provided by the user (SNAPP analysis results, input sequences and parameters). This could be done by rearranging the elements so the blue arrows in the grey box form a linear workflow, I also suggest removing the grey box.
(ps. R2 notes 'Figure 2 is really hectic' - it is possible that was a typo and they actually meant Figure 1 here - but see below for my own suggestion re Figure 2).


5. For general use, some indication as to the minimum set of SNAPP results that are sufficient for analysis would be helpful.In line 136-138 you describe a sampling protocol - but is it sufficient for any size sequence set ? It is important to clearly state any caveats here since users will typically adopt the same settings settings when using the package.
i. R2 suggests provision of an effect size statistic - possibly as further work.


6. Expanded presentation & analysis of results.
i. Figure 2 is very difficult to interpret. A simpler way to quantify classification performance is Matthews Correlation coefficient. It may also allow trends to be identified for different metrics as the degree of gene flow varies.
ii. line 205: "(52 out of 100 gene flow datasets; range 0.0000 – 0.0028)" - this is not at all sufficient for readers to evaluate the performance of the method and it is not clear what the 0.00-0.0028 range refers to - (presumably, RF distances ?). At least show results in tabular form to allow all details to be inspected, and ideally present a plot in order to explore (as reviewers suggest) whether there is any correlation between the variation and/or reliability of one or more metrics with different degrees of gene flow.

iii. line 202-204. "Our results identified one statistic that performed well. The standard deviation of tree likelihoods correctly classified 83% of simulated datasets, with 14% false negatives and 3% false positives. ". R1 in particular highlights it would be useful to propose a reason as to why this statistic is the most effective discriminator. I can provide no reference at this moment but I consider this to be a known effect: a narrow variance in quality metrics computed for solutions such as phylogenetic trees are indicative of good convergence (as opposed to observing poorly converged sets due to poor parameterisation such as model violations).

iv. line 225-228. R1 provides potential alternative approaches that could also be considered here. Personally I found the 'we do not advocate' but nonetheless 'research is still warranted' argument here too complex to follow on first reading. (Part of this was indeed due to my own confusion regarding Figure 2 and lack of experience with the Fst metric)

7. R Packaging/publication. R2 notes that the package does not cleanly install in the most recent version of R. Reviewer 3 in particular suggests an example script would help. Highlighting its presence in the text may help others, but I also note here that the vignette in the git repository ( https://raw.githubusercontent.com/P2C2M/P2C2M_SNAPP/master/vignettes/using_p2c2m-snapp.Rmd ) does not appear to have been picked up by R package repository web sites such as https://rdrr.io/github/P2C2M/P2C2M_SNAPP/ - indicating there may be an issue with the package's structure.


8. Typographic revisions

line 85-87. The R Core Team citation is for R - so it should be after 'implemented as an R Package'.

line 230. "However, if it occurs" - replace 'it' with 'gene flow'.

Reviewer 1 ·

Basic reporting

I enjoyed reading this manuscript, it is well-written, succinct, and clearly describes some new, important tools that were developed for assessing the reliability of phylogenetic inferences under coalescent models.

Experimental design

The methods are well documented and suited to addressing the questions presented here.

Line 91: A minor point, but would it be possible to add an option for users to sample uniformly from the posterior? This might be preferred in some cases to reduce sample autocorrelation.

As the authors point out, poor performance of the Fst summary statistic is unfortunate because its calculation only requires the use of the raw datasets, and thus may be more computationally approachable for some users. Although it’s not necessary, calculating several other population genetics summary statistics from the posterior and posterior predictive datasets seems like a relatively straightforward extension, and if any were found to be useful, I would guess that the tests may be more widely applied. Another option that comes to mind would be to develop a test focused around concordance factors calculated from the SNP datasets.

Validity of the findings

no comment

Additional comments

This manuscript documents an important area of research, detecting model violations in empirical data is foundational to accurate phylogenetic inference. Generally, I think the manuscript is in great shape, though I’ll provide a few suggestions for the authors consideration:

The extensive focus on gene flow as a violation of the MSCM in the manuscript felt limited in scope. While I recognize that was the focus of the simulation testing in the manuscript, in theory, the test statistics identified here could be useful in detecting poor fit between the model and a dataset, regardless of the processes that produce this. Some discussion of this issue would be nice.

I’d imagine the poor performance of many of the statistics that were evaluated was somewhat unsatisfying to the authors as well. Some further discussion of this may be warranted, especially for those test statistics with high false positive rates, as posterior predictive checks are often thought to be highly conservative. Do the authors have any idea why the MLSD is so useful for detecting poor fit in this scenario compared to the others that are evaluated? Could it be related to the specific simulation conditions that were considered? Although quite a bit of work has been done recently developing methods for assessing model fit in phylogenetics, less work has been directed towards identifying when and why some test statistics work better than others. Though outside the scope of this manuscript, I’m sure potential users of these tools and other workers in the field would appreciate any ideas about this the authors have gleaned from their work.

Finally, I think some further presentation of the results would be useful. For example, the manuscript doesn’t report the performance of the summary statistics under different levels of gene flow, despite the simulations being done under a very large range. One might expect high rates of gene flow to be more problematic for phylogenetic inference and easier to detect using tests of model fit. Was there any correlation between the migration rates under which the data was simulated and the test statistic values? If there were, Fig. 2 might warrant some modification to demonstrate this.

Reviewer 2 ·

Basic reporting

The manuscript is well written, clear and concise.

General Comments:
1). This is personal preference, but I find figure 2 really hectic and lacking in detail. Since the SNAPP parts of the analyses aren't talked about in the manuscript at all, and aren't controlled by the R package in review, perhaps minimizing those parts of the figure, and expanding the "grey box" portion to add more detail in what happens in each step would be a more clear way to show the work flow?

Minor Comments:

Line 44: occurs should be occur

Experimental design

General Comments:

1). The package doesn't seem to install with R version >3.6. That should be listed in the manuscript someplace as minimum system requirement, or perhaps is a bug in the current CRAN archive? If it's intended, the manuscript should formally define those system requirements at some point.

Validity of the findings

1.) The findings of which test statistics have a propensity to report false negatives is quite interesting. This is an area were PPS needs far more work. It would be helpful, almost necessary in some ways, for the package to include some type of effect size statistic. Without it, detecting model violation is a bit of a binary result. Yes there is model violation, or no there is not, without any information helping a researcher determine what the impact of that violation is on their analyses. One possible example is in the Brown et. al. paper the authors cite, though I'm not sure if it would be appropriate for this package, something similar would be very helpful.

Additional comments

Overall the work is solid and will add another needed piece to researchers using PPS to determine the model adequacy of their experimental design. Incorporating this package in an already available R package in a familiar framework will hopefully help users become more accustomed to doing this type of model checking as a natural part of their workflow.

·

Basic reporting

Overall, I find this manuscript well written and the present application useful for the community. As you will see in my comments, my main concerns regard the clarity of the manuscript. If found that it could have been more detailed in several parts.


General comments:
It would be very helpful if you could provide a better overview about the summary statistics and methods used. I'ld suggest to add a box with the brief definition of specific summary statistics (e.g. Fst, RF and KF tree distance).

The interactions of the user was not very clear. When does the user need to run an analysis manually (e.g., running SNAPP or computing summary stats) and when does this happen automatically? Would it be possible to insert a code or pseudo-code example with the steps a user has to take for the analysis pipeline to run?

Could you add a sketch showing the two models used in the simulation. I find it easier to see the gene-flow example instead of only reading the description.

What was the motivation to choose these specific summary statistics? Are there any other summary statistics in the literature that could be used too? How easy would it be to add new summary statistics?

It took me quite some time to wrap my head around Figure 2 and the results. Perhaps it would be better to split the results into two plots; one for each simulation setting. Some of the results are very surprising. Why are the two tree metrics (RF and KF) always rejecting one model? Why is it that a summary statistic can have so many false positives? I would love to see some insights here.




Specific comments:
What doesP2C2M actually stand for? Maybe you could provide the full name when you first use it (line 77).

It was not really clear to me that you were using both inference based and data based summary statistics. Perhaps you could make this more clear in the introduction (paragraph starting at line 66) and in the section called "Pipeline".

In lines 73 to 75 you are stating that there should should be little difference in the summary statistics if the model provides a good fit. This statements is technically not correct. Let us consider a hypothetical example as follows with two models A and B. Model A produces posterior predictive datasets where the summary statistics are very, very close to the ones computed from the empirical data, but always slightly larger. Thus, the posterior predictive p-value is 0 (rejected). Model B produces datasets with widely spread summary statistics (many of them not close to the empirical summary statistic), but half of the values are smaller and half of the values are larger than the empirical value. Thus, the p-value would be 0.5 (not rejected). So it is not (only) about closeness of the simulated datasets but more importantly that the empirical dataset is not unexpected because it lies in the simulated range of the model.

What do you mean by "samples at random" (line 91)? Wouldn't it be actually more efficient if you take every k-th sample because then you minimize the autocorrelation of MCMC samples (standard thinning)?

Please add a link to the tutorials in line 102.

Replace the "with" with "have" in line 109.

In your definition of p-values, what do you do when you have exactly the same value for the summary statistic? I believe this might happen very often for the RF tree distance. In our work, we specifically defined the lower p-value as samples being smaller or equal, thus using a conservative approach (see Höhna et al. 2018)

The sentence starting at line 144 can be removed; it is a repetition of line 120.

In line 165 you should provide the full link to the GitHub repository.

What is a gamma tree prior (line 165)? Commonly we use a birth-death process as a species tree prior, or the MSC as a gene tree prior. Did you mean a gamma prior on the population size?

What gamma rate prior do you mean (line 166)? On which rate is this prior used? Substitution rate, clock rate, among site rate variation?

What specifically do you mean by "the effect it could have" (line 235)? Could you provide an example what you would researchers suggest to write if they find model violations using P2C2M.SNAPP?

Experimental design

The experimental design is well suited. I have only some concerns about sufficient details about the methods, as provided above.

Validity of the findings

The results could be discussed a bit more. Overall, the results of the comparison between summary statistics was very brief and should be expanded. Clearly some results are surprising, and it would be nice to see what the next steps could be.

Additional comments

I reviewed the manuscript with a first year graduate student in my group. I've included her feedback in my evaluation. She strongly thought that the manuscript could be more precise with much more details about specific steps in the analysis pipeline. I agree that the present manuscript requires a good background knowledge in field and is hard to fully understand without. I leave it up to the authors if they prefer to refer to literature or expand the present manuscript to make it more accessible to young scientists.

---

## Round 0.2 · accepted · Accept

Thanks for addressing the comments from the previous round. One reviewer has provided a few suggestions and typographical revisions; please also consider the comments in my annotated PDF.

Reviewer 1 ·

Basic reporting

no comment

Experimental design

no comment

Validity of the findings

no comment

Additional comments

This manuscript reports progress on an important series of tools for detecting model violations in empirical data analyzed under the multispecies coalescent. I thank the authors for addressing my suggestions in my previous review of the study, and think that the manuscript (which was already in good shape) is clear, succinct, and a valuable contribution.

A couple minor comments that came up upon my reading that the authors may want to consider addressing in the manuscript:

Line 44: Should be “e.g.” not “i.e.”?

Line 99: Is there a justification or citation that can be provided for the minimum of 100 trees?

Fst outlier test: I am assuming users have the ability to control the parameters that determine the sensitivity of these tests to detecting outliers? Do the authors have any data (or even intuition) on how doing so might impact the performance of the posterior predictive test results?

Results: Did the authors qualitatively look at the effect of altering the p-value threshold (e.g., using 0.01 instead of 0.05) for model violations? I wonder if this would have any effect on the number of type 1 and type 2 errors for different tests.